# Synthesis, Chiral Resolution and Enantiomers Absolute Configuration of 4-Nitropropranolol and 7-Nitropropranolol

**DOI:** 10.3390/molecules28010057

**Published:** 2022-12-21

**Authors:** Rosa Sparaco, Antonia Scognamiglio, Angela Corvino, Giuseppe Caliendo, Ferdinando Fiorino, Elisa Magli, Elisa Perissutti, Vincenzo Santagada, Beatrice Severino, Paolo Luciano, Marcello Casertano, Anna Aiello, Gilberto De Nucci, Francesco Frecentese

**Affiliations:** 1Department of Pharmacy, University of Naples Federico II, Via D. Montesano 49, 80131 Naples, Italy; 2Department of Public Health, University of Naples Federico II, Via Pansini 5, 80131 Naples, Italy; 3Department of Pharmacology, Faculty of Medical Sciences, State University of Campinas (UNICAMP), Campinas 13083-970, SP, Brazil

**Keywords:** 4-nitropropranolol, 7-nitropropranolol, chiral HPLC, NMR absolute configuration assignment, Riguera’s method

## Abstract

We recently identified 6-nitrodopamine and other nitro-catecholamines (6-nitrodopa, 6-nitroadrenaline), indicating that the endothelium has the ability to nitrate the classical catecholamines (dopamine, noradrenaline, and adrenaline). In order to investigate whether drugs could be subject to the same nitration process, we synthesized 4-nitro- and 7-nitropropranolol as probes to evaluate the possible nitration of the propranolol by the endothelium. The separation of the enantiomers in very high yields and excellent enantiopurity was achieved by chiral HPLC. Finally, we used Riguera’s method to determine the absolute configuration of the enantiomers, through double derivatization with MPA and NMR studies.

## 1. Introduction

Propranolol is a beta blocker used to treat cardiovascular disorders, tremors, hypertension, and angina. Propranolol also is an effective and safe drug for treating migraine headaches, anxiety disorders, and infantile hemangiomas. Moreover, several studies have recently reported that patients receiving propranolol reduced their risk of neck and head, colon, stomach, and prostate cancers [1,2].

Nitroaromatic compounds are relatively rare in nature and have been introduced into the environment mainly by human activities. The nitro group provides chemical and functional diversity in these molecules due to its electron-withdrawing nature that, in concert with the stability of the benzene ring, makes nitroaromatic compounds resistant to oxidative degradation [3].

6-Nitrocatecholamines are a wide family of neurotransmitters; 6-nitroepinephrine and 6-nitrodopamine belong to this family and have been shown to have several biological activities, such as inhibition of neuronal norepinephrine uptake, catechol-O-methyltransferase activity, neuronal nitric oxide synthase activity, and contractile capacity at vascular adrenoceptors [4]. 

6-Nitrodopamine, in particular, has been recently identified by us from vascular tissues, such as human umbilical vessels [5], and its release proved to be substantially reduced when the endothelium was mechanically removed from these vessels. 

6-Nitrodopamine is one hundred times more potent than dopamine, noradrenaline, and adrenaline as a positive chronotropic agent [6]. The basal release of 6-nitrodopamine was detected in *Chelonoidis carbonarius* aortic rings [7], as well as in rat [8] and human vas deferens, too [9].

We have now identified other catecholamines (6-nitrodopa and 6-nitroadrenaline), indicating that the endothelium has the ability to nitrate the classical catecholamines (dopamine, noradrenaline, and adrenaline). Thus, it is very possible that not only endogenous compounds but also drugs could be subject to the same process. 

Propranolol belongs to the 3-(aryloxy)-1-(alkylamino)-β-blocker family, a class of molecules with activity residing mainly in the *S* isomers [10]. Indeed, (*S*)-(−)-propranolol is 98 times more active than (*R)*-(+)-enantiomer. Methods reported for the synthesis of (*S*)-propranolol have been reported in the literature, ranging from the resolution of a chiral mixture of a final compound, or of intermediates, to asymmetric synthesis [11,12,13,14,15]. Moreover, several examples of propranolol derivatives, designed and synthesized for variable purposes, have been reported in the literature [16,17,18,19].

In order to investigate whether drugs could be subject to the same nitration process discovered for endogenous catecholamines, we synthesized 4-nitro- and 7-nitropropranolol. These propranolol derivatives will be used as probes to evaluate the possible nitration of the propranolol by the endothelium. Because enantiomers may not share the same pharmacologic profile, in order to discover differences in nitration by the endothelium of different enantiomers of the same compound, we focused our attention on chiral resolution.

The pure enantiomers were obtained by chiral chromatography, and the absolute configuration of secondary alcohol was obtained by applying Riguera’s method based on the sign distribution of Δ*δ*_H_ [20,21].

## 2. Results

Our study commenced with the preparation of (±)-4-nitropropranolol (**2a**) and (±)-7-nitropropranolol (**2b**).

The nitrate derivatives of propranolol were obtained following the procedure described in Figure 1.

In particular, (±)-4-nitropropranolol (**2a**) and (±)-7-nitropropranolol (**2b**) were synthesized from the reaction between 4-nitro-1-naphthol or 7-nitro-1-naphthol and epichlorohydrin; the glycidyl naphthyl ethers (**1a** and **1b**) thus obtained were reacted with isopropylamine providing the racemic mixtures of the nitrate derivatives of propranolol.

The four enantiomers (two for each racemic mixture) were then obtained by resolution of the corresponding racemic mixture by chiral HPLC. Chromatograms of preparative purifications are reported in Figure 1 (Panels A and B for 4-nitropropranolol and 7-nitropropranolol, respectively).

At the end of racemic resolutions, each enantiomer was further analyzed to determine the purity, which was generally >98% with an enantiomeric excess >98%.

The purity of (+) and (−) enantiomers was assessed by chiral HPLC with the same solvents and conditions as for preparative purification, but with a flow rate of 1 mL/min. The chromatograms obtained for each single enantiomer are reported in the Appendix A.

The optical rotation values for each enantiomer were then determined by optical polarimetry.

The isomers corresponding to Peak 1 gave positive [α]_D_ values for both 4-nitro- and for 7-nitropropranolol. The optical evaluation of a solution of each isomer corresponding to Peak 2 confirmed that these were characterized by negative [α]_D_ values. 

Finally, the absolute configuration (AC) of the synthesized compounds was determined by the double derivatization of each chiral compound with an adequate chiral derivatizing agent represented by α-methoxyphenylacetic acid (MPA) followed by an NMR analysis of the resulting derivatives. 

Defining the absolute stereochemistry is essential to understanding many of the chemical, biological, and pharmaceutical properties of new chiral compounds, and therefore developing easy-to-use methods for its determination in solution is of utmost interest for researchers involved in bioactive compounds. Nowadays, the method of choice for the AC assignment to chiral secondary alcohols by NMR is Mosher’s method, first described in 1973, and since then, it has become increasingly popular, thanks to its demonstrated reliability [22]. The method has been extensively studied and further optimized by Riguera’s research group [20].

Each optically pure derivative (+)-4-nitropropranolol, (−)-4-nitropropranolol, (+)-7-nitropropranolol, and (−)-7-nitropropranolol), respectively named as (+)-**2a**, (−)-**2a,** (+)-**2b,** and (−)-**2b,** was reacted both with (*R*)-(−)-α-methoxyphenylacetic acid and (*R*)-MPA and with (*S*)-(+)-α-methoxyphenylacetic acid and (*S*)-MPA, which has been proven to be a chiral derivatizing agent (CDA) of choice for alcohols [20,21].

The synthetic procedure is reported in Figure 2. 

The assignment of the absolute configuration of secondary alcohol was based on the sign distribution of Δ*δ*_H_ (calculated as *δ*_H_^R^ − *δ*_H_^S^), obtained from the comparison of the NMR spectra of the different MPA-derived compounds. In that regard, the NMR signals were identified, paying special attention to the protons located at the two substituents of the chiral carbon, which constitute the L1 and L2 groups (Figures 4, 7, 10, and 13). We interpreted the experimental Δ*δ^RS^* data as the result of the shielding/deshielding anisotropic effects exerted by the MPA units around the stereogenic center, which showed a different and opposite behavior when (*R*)- or (*S*)-MPA was used. This effect is moved in consistent chemical shift differences that allowed to analyze the Δ*δ*^RS^ pattern according to Riguera’s proposed model [20,21]. Indeed, the analysis of the Δ*δ*^RS^ (*δ*^R^ − *δ*^S^) parameters, obtained from the comparison of the spectra, showed a consistent distribution of positive and negative Δ*δ* values around the stereogenic carbon and allowed us to deduce:-the *S* configuration for (+)-4-NO_2_-propranolol and (+)-7-NO_2_-propranolol;-the *R* configuration for (−)-4-NO_2_-propranolol and (−)-7-NO_2_-propranolol.

## 3. Materials and Methods

### 3.1. Chemistry

All commercial reagents and solvents were purchased from Merck(Darmstadt, Germany), TCI Europe (Zwijndrecht, Belgium) and Enamine (Kyiv, Ukraine). Reactions were stirred at 400 rpm by a Heidolph MR Hei-Standard magnetic stirrer. Solutions were concentrated with a Buchi R-114 rotary evaporator (Flawil, Switzerland) at low pressure. All reactions were followed by TLC carried out on Merck silica gel 60 F254 plates with a fluorescent indicator on the plates and were visualized with UV light (254 nm). Preparative chromatographic purifications were performed using silica gel column (Kieselgel 60). Chiral HPLC resolution was performed using a WATERS (Milford, MA, USA) Quaternary Gradient Mobile 2535 instrument equipped with WATERS UV/Visible Detector 2489 set to a dual-wavelength UV detection at 254 and 280 nm. The chiral resolutions were achieved on the Kromasil 5-Amycoat column Phenomenex (150 mm × 21.2 mm, 5 μm particle size). 

LC-MS analysis was performed using a VANQUISH FLEX module comprising a quaternary pump with degasser, an autosampler, a column oven (set at 40 °C), a diode-array detector DAD, and a column as specified in the respective methods below. The MS detector (ISQ Thermo Fisher Scientific, Waltham, MA, USA) was configured with an electrospray ionization source. Mass spectra were acquired by scanning from 100 to 700 in 0.2 s. The capillary needle voltage was 3 kV in positive and 2 kV in negative ionization mode, and the source temperature was maintained at 250 °C. Nitrogen was used as the nebulizer gas. Reversed phase UHPLC was carried out on a Luna Omega-C18 column Phenomenex (3 µm, 50 × 2.1 mm) with a flow rate of 0.600 mL/min. Two mobile phases were used, mobile phase A: water with 0.1% formic acid and mobile phase B: acetonitrile (LiChrosolv for LC-MS Merck), and they were employed to run gradient conditions from 15% B for 0.20 min, from 10% to 95% for 1.60 min, 95% B for 0.80 min, and 15% B for 0.10 min, and these conditions were held for 1.05 min in order to re-equilibrate the column (Total Run Time 3.55 min). An injection volume of 0.8 µL was used. Data acquisition was performed with Chromeleon 7. HPLC on silica gel was carried out on a Luna SiO_2_ column (Phenomenex, 3 µM, 50 × 4.6 mm) using a Knauer K-501 apparatus equipped with a Knauer K-2301 RI detector (LabService Analytica s.r.l., Anzola dell’Emilia, Italy). Melting points, determined using a Buchi Melting Point B-540 instrument, are uncorrected and represent values obtained on recrystallized or chromatographically purified material. Mass spectra of final products were performed on LTQ Orbitrap XL™ Fourier transform mass spectrometer (FTMS) equipped with an ESI ION MAX™ (Thermo Fisher Scientific, Waltham, MA, USA) source operating in positive mode. MeOH was used as solvent for compound infusion into LTQ Orbitrap source. NMR experiments were recorded on Varian Mercury Plus 700 MHz instrument. All spectra were recorded in CDCl_3_. Chemical shifts were reported in ppm and referenced to the residual solvent signal (CDCl_3_: *δ*_H_ = 7.26, *δ*_C_ = 77.0). The following abbreviations are used to describe peak patterns when appropriate: s (singlet), d (doublet), dd (double doublet), t (triplet), m (multiplet), and bs (broad singlet). NMR signals and monodimensional and bidimensional NMR spectra of 4-nitropropranolol **2a** and 7-nitropropranolol **2b** are reported in Appendix A. NMR signals, ^1^H-NMR spectra, ^13^C-NMR spectra, and HRMS spectra for bis-MPA derivatives of 4-nitropropranolol and 7-nitropropranolol are reported in the Appendix A. Optical rotation values were determined using a JACSCO P-2000 series polarimeter (Tokyo, Japan).

#### 3.1.1. 2-(((4-Nitronaphthalen-1-yl)oxy)methyl)oxirane (**1a**)

4-Nitro-1-naphthol (0.6 g, 3.18 mmol) was dissolved in 10 mL of ethanol:water (9:1, *v*/*v*) followed by adding 0.18 g of KOH into the mixture under stirring for 30 min. Successively, 1 mL of epichlorohydrin (12.7 mmol) was added until the color of the mixture changed to orange-yellow. The mixture was heated under reflux for 4 h, and the formation of the intermediate **1a** was monitored by TLC (dichloromethane). Then, the solvent of the reaction mixture was evaporated, and the residue was taken up in dichloromethane (90 mL) extracted with brine (3 × 30mL). The organic layer was dried on anhydrous Na_2_SO_4_, concentrated, and chromatographed on a silica gel column (dichloromethane) to provide the intermediate **1a** as a yellow powder (298 mg). Yield: 38%. m.p. 80–81 °C. ^1^H NMR (CDCl_3_) *δ* 8.75 (d, J = 8.6, 1H), 8.41 (d, J = 8.6, 1H), 8.35 (d, J = 8.2, 1H), 7.74 (t, J = 8.6, 1H), 7.60 (t, J = 8.6, 1H), 6.81 (d, J = 8.6, 1H), 4.56 (dd, J = 11.1, 1.9, 1H), 4.18 (dd, J = 11.1, 6.0, 1H), 3.53 (m, 1H), 3.01 (t, J = 4.0, 1H), and 2.86 (dd, J = 4.0, 2.6, 1H). ^13^C NMR (CDCl_3_) *δ* 44.5, 49.7, 69.8, 102.8, 122.8, 123.5, 125.6, 126.6, 126.7, 126.9, 130.1, 139.7, and 159.2. ESI-MS [M + H]^+^ *m*/*z* calc. 245.23 for C_13_H_11_NO_4_ found 246.2.

#### 3.1.2. 2-(((7-Nitronaphthalen-1-yl)oxy)methyl)oxirane (**1b**)

Compound **1b** was obtained following the same procedure reported for **1a** starting from 7-nitro-1- naphthol (0.6 g, 3.18 mmol). Yellow powder. 430 mg. Yield: 55%. m.p. 103–104 °C. 1H NMR (CDCl_3_) *δ* 9.23 (d, J = 2.0, 1H), 8.23 (dd, J = 9.0, 2.0, 1H), 7.89 (d, J = 9.0, 1H), 7.58 (t, J = 8.0, 1H), 7.51 (d, J = 8.2, 1H), 6.95 (d, J = 7.7, 1H), 4.46 (dd, J = 10.9, 3.0, 1H), 4.18 (dd, J = 10.9, 5.9, 1H), 3.53 (m, 1H), 3.03 (t, J = 4.4, 1H), and 2.85 (dd, J = 4.4, 2.6, 1H). ^13^C NMR (CDCl_3_) *δ* 44.8, 50.0, 69.6, 106.6, 119.7, 120.0, 120.5, 124.3, 129.5, 130.2, 136.9, 145.1, and 155.7. ESI-MS [M + H]^+^ *m*/*z* calc. 245.23 for C_13_H_11_NO_4_ found 246.2.

#### 3.1.3. (±)-1-(Isopropylamino)-3-((4-nitronaphthalen-1-yl)oxy)propan-2-ol (**2a**)

The glycidyl naphthyl ether **1a** (0.25 g, 1.02 mmol) was dissolved in 30 mL of isopropylamine, and the stirred mixture was heated to 30 °C for 1 h. Then, the solvent of the reaction mixture was evaporated, and the residue was chromatographed on a silica gel column (dichloromethane/methanol 8:2, *v*/*v*) to provide ±-4-nitropropranolol **2a** as a yellow powder (85 mg). Yield: 28%. m.p. 98–99 °C. ^1^H NMR (CDCl_3_) *δ* 8.78 (d, J = 8.6, 1H), 8.38 (d, J = 8.6, 1H), 8.36 (d, J = 8.2, 1H), 7.74 (t, J = 8.6, 1H), 7.60 (t, J = 8.6, 1H), 6.84 (d, J = 8.6, 1H), 4.28 (m, 1H), 4.24 (m, 1H), 4.17 (m, 1H), 3.03 (dd, J = 12.2, 4.0, 1H), 2.87–2.83 (m, overlapped, 2H), 1.12 (d, J = 3.4, 3H), and 1.11 (d, J = 3.4, 3H). ^13^C NMR (CDCl_3_) *δ* 23.1, 23.2, 49.0, 49.1, 68.3, 71.4, 102.8, 122.6, 123.5, 125.5, 126.5, 126.9, 127.0, 130.1, 139.4, and 159.5. ESI-MS [M + H]^+^ *m*/*z* calc. 304.34 for C_16_H_20_N_2_O_4_ found 305.2.

#### 3.1.4. (±)-1-(Isopropylamino)-3-((7-nitronaphthalen-1-yl)oxy)propan-2-ol (**2b**)

Compound **2b** was obtained following the same procedure reported for **2a** starting from glycidyl naphthyl ether **1b** (0.25 g, 1.02 mmol). Yellow powder. 96 mg. Yield: 31%. m.p. 139–140 °C. ^1^H NMR (CDCl_3_) *δ* 9.20 (d, J = 2.0, 1H), 8.22 (dd, J = 9.0, 2.0, 1H), 7.88 (d, J = 9.0, 1H), 7.58 (t, J = 8.0, 1H), 7.49 (d, J = 8.2, 1H), 6.96 (d, J = 7.7, 1H), 4.24–4.19 (m, overlapped, 2H), 4.18 (m, overlapped, 1H), 3.03 (dd, J = 12.3, 3.6, 1H), 2.88–2.85 (m, overlapped, 2H), 1.13 (d, J = 3.4, 3H), and 1.12 (d, J = 3.4, 3H). ^13^C NMR (CDCl_3_) *δ* 23.0, 23.3, 48.9, 49.3, 68.1, 71.1, 106.5, 119.6, 119.9, 120.2, 124.3, 129.0, 136.8, and 156.0. ESI-MS [M + H]^+^ *m*/*z* calc. 304.34 for C_16_H_20_N_2_O_4_ found 305.2.

### 3.2. Chiral Resolution

Resolution of the racemic mixtures containing (±)-4-nitropropranolol **2a** or (±)-7-nitropropranolol **2b** was performed using a Waters 2535 Quaternary Gradient Mobile equipped with Waters 2489 UV/Visible Detector set to a dual-wavelength UV detection at 254 and 280 nm. The chiral resolutions were achieved on the Kromasil 5-Amycoat column Phenomenex (150 mm × 21.2 mm, 5 μm particle size). Two mobile phases, in isocratic condition, were used. Mobile phase A: n-Hexane (Chromasolv Sigma-Aldrich) and mobile phase B: Isopropanol (Chromasolv Sigma-Aldrich) + 0.1% Diethylamine. Ratio of the mobile phases was 86(A):14(B). The racemic mixtures were dissolved in ethanol at concentration of 75 mg/mL, and the injection volume was 200 μL (repeated 5 times); the sample was eluted from the column at a flow rate of 15.0 mL/min at room temperature (pressure: ≈500 psi). At the end of racemic resolutions, 30 mg of each enantiomer was collected. Purity of (+) and (−) enantiomers was assessed by chiral HPLC using the Kromasil 5-Amycoat column Phenomenex (150 mm × 4.6 mm, 5 μm particle size) with the same solvents and conditions as for preparative purification, but with a flow rate of 1 mL/min.

### 3.3. Determination of Optical Rotation Values

Optical rotation values (Table 1) were determined in a JASCO P-2000 series polarimeter for each single enantiomer. 

### 3.4. Absolute Configuration Assignment

General synthetic procedure for the obtaining of the bis MPA derivatives of (±)-4-nitropropranolol (**2a**) and (±)-7-nitropropranolol (**2b**):

Each pure enantiomer was converted into the relevant MPA derivative by applying the double derivatization method proposed by Riguera and coworkers, which is based on the use of both *R* and *S* enantiomer of the α-methoxyphenylacetic acid [10,11]. In detail, a portion of the pure nitro propranolol compounds ((+)-**2a**, (−)-**2a**, (+)-**2b**, and (−)-**2b**) was dissolved in dry dichloromethane (1mL). To this solution, 2.2 equivalent (eq.) of either (*R*)-(−)-α-methoxyphenylacetic acid or (*S*)-(+)-α-methoxyphenylacetic acid, 2.2 eq. of 1-ethyl-3-(3-dimethylaminopropyl)carbodiimide (EDC), and a catalytic amount of DMAP were added. The mixture was stirred overnight at room temperature (rt) before removing the solvent under vacuum. Each residue was separately purified by HPLC on silica gel (Luna 3-µM SiO_2_ column, flow rate 0.5 mL/min, mobile phase n-hexane/ethyl acetate 6:4 *v*/*v*) providing the desired compounds in pure form. The reaction conditions and yields are summarized in Table 2. ^1^H NMR and ^13^C NMR shifts were assigned by HSQC and HMBC experiments and are reported in Figure 2, Figure 3, Figure 4, Figure 5, Figure 6, Figure 7, Figure 8, Figure 9, Figure 10, Figure 11, Figure 12 and Figure 13.

Bis-(*R*)-MPA derivative of (+)-4-NO_2_-propranolol: the analysis of the key correlations ^1^H-^13^C (^2,3^ J) recorded by gradient 2D NMR experiments allowed to assign all proton and carbon resonances for the synthesized compound (Figure 2). HRESI-MS: *m*/*z* 601.2541 [M + H]^+^ (calculated for C_34_H_37_O_8_N_2_: 601.2544); HRESI-MS *m*/*z* 623.2359 [M+Na]^+^ (calculated for C_34_H_36_O_8_N_2_Na: 623.2364, Appendix A).

Bis-(*S*)-MPA derivative of (+)-4-NO_2_-propranolol: all proton and carbon resonances for the synthesized compound are reported in Figure 3. HRESI-MS: *m*/*z* 601.2549 [M + H]^+^ (calculated for C_34_H_37_O_8_N_2_: 601.2544, Appendix A).

Applying Riguera’s method [20,21], by double derivatization procedure, for assigning the absolute configuration of secondary alcohol based on the sign distribution of Δ*δ*_H_ (calculated as *δ*_H_^R^ − *δ*_H_^S^) allowed assigning *S* configuration to the (+)-4-NO_2_-propranolol (Figure 4).

Bis-(*R*)-MPA derivative of (+)-7-NO_2_-propranolol: the analysis of the key correlations ^1^H-^13^C (^2,3^ J) recorded by gradient 2D NMR experiments allowed to assign all proton and carbon resonances for the synthesized compound (Figure 5). HRESI-MS: *m*/*z* 601.2539 [M + H]^+^ (calculated for C_34_H_37_O_8_N_2_: 601.2544); HRESI-MS *m*/*z* 623.2358 [M+Na]^+^ (calculated for C_34_H_36_O_8_N_2_Na: 623.2364, Appendix A).

Bis-(*S*)-MPA derivative of (+)-7-NO_2_-propranolol: all proton and carbon resonances for the synthesized compound are reported in Figure 6. HRESI-MS: *m*/*z* 601.2550 [M + H]^+^ (calculated for C_34_H_37_O_8_N_2_: 601.2544); HRESI-MS *m*/*z* 623.2368 [M+Na]^+^ (calculated for C_34_H_36_O_8_N_2_Na: 623.2364, Appendix A).

Applying Riguera’s method [20,21], by double derivatization procedure, for assigning the absolute configuration of secondary alcohol based on the sign distribution of Δ*δ*_H_ (calculated as *δ*_H_^R^ − *δ*_H_^S^) allowed assigning *S* configuration to the (+)-7-NO_2_-propranolol (Figure 7).

Bis-(*R*)-MPA derivative of (−)-4-NO_2_-propranolol: the analysis of the key correlations ^1^H-^13^C (^2,3^ J) recorded by gradient 2D NMR experiments allowed to assign all proton and carbon resonances for the synthesized compound (Figure 8). HRESI-MS: *m*/*z* 601.2537 [M + H]^+^ (calculated for C_34_H_37_O_8_N_2_: 601.2544); HRESI-MS *m*/*z* 623.2356 [M+Na]^+^ (calculated for C_34_H_36_O_8_N_2_Na: 623.2364, Appendix A).

Bis-(*S*)-MPA derivative of (−)-4-NO_2_-propranolol: the analysis of the key correlations ^1^H-^13^C (^2,3^ J) recorded by gradient 2D NMR experiments allowed to assign all proton and carbon resonances for the synthesized compound (Figure 9). HRESI-MS *m*/*z* 623.2365 [M+Na]^+^ (calculated for C_34_H_36_O_8_N_2_Na: 623.2364, Appendix A).

Applying Riguera’s method [20,21], by double derivatization procedure, for assigning the absolute configuration of secondary alcohol based on the sign distribution of Δ*δ*_H_ (calculated as *δ*_H_^R^ − *δ*_H_^S^) allowed assigning ***R*** configuration to the (−)-4-NO_2_-propranolol (Figure 10).

Bis-(*R*)-MPA derivative of (−)-7-NO_2_-propranolol: the analysis of the key correlations ^1^H-^13^C (^2,3^ J) recorded by gradient 2D NMR experiments allowed to assign all proton and carbon resonances for the synthesized compound (Figure 11). HRESI-MS: *m*/*z* 601.2542 [M + H]^+^ (calculated for C_34_H_37_O_8_N_2_: 601.2544); HRESI-MS *m*/*z* 623.2360 [M+Na]^+^ (calculated for C_34_H_36_O_8_N_2_Na: 623.2364, Appendix A).

Bis-(*S*)-MPA derivative of (−)-7-NO_2_-propranolol: the analysis of the key correlations ^1^H-^13^C (^2,3^ J) recorded by gradient 2D NMR experiments allowed to assign all proton and carbon resonances for the synthesized compound (Figure 12). HRESI-MS: *m*/*z* 601.2542 [M + H]^+^ (calculated for C_34_H_37_O_8_N_2_: 601.2544); HRESI-MS *m*/*z* 623.2361 [M+Na]^+^ (calculated for C_34_H_36_O_8_N_2_Na: 623.2364, Appendix A).

Applying Riguera’s method [20,21], by double derivatization procedure, for assigning the absolute configuration of secondary alcohol based on the sign distribution of Δ*δ*_H_ (calculated as *δ*_H_^R^ − *δ*_H_^S^) allowed assigning *R* configuration to the (−)-7-NO_2_-propranolol (Figure 13).

## 4. Conclusions

In summary, we have reported a practical synthesis of 4-nitro- and 7-nitropropranolol as racemic mixtures. Moreover, we disclosed a very efficient chiral HPLC method for separating the enantiomers that allowed very high yields and enantiopurity. Finally, we used Riguera’s method to determine the absolute configuration of the enantiomers through double derivatization with MPA and NMR studies. 

We believe that these synthetic procedures will be useful for preparing various enantiopure biologically active products and drugs containing chiral secondary alcohols.

## Data Availability

Not applicable.

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
