# Peer review of "Synthesis, Chiral Resolution and Enantiomers Absolute Configuration of 4-Nitropropranolol and 7-Nitropropranolol"

_molecules, 2022, doi:10.3390/molecules28010057_

Round 1
Reviewer 1 Report
Recommendation
Reject
Comments:
This manuscript describes the synthesis of 4-nitro and 7-nitropropranolol as racemic mixtures and isolation of the enantiomers employing chiral HPLC method. The absolute configurations of the enantiomers were determined by double derivatization with MPA and NMR studies according to Riguera’s method. This research, however, does not have any novelty or significance in the design, synthesis, and resolution of chiral compounds. The results are predictable with no surprise. The characterization and analysis of the data are facile and straightforward. Consequently, readers can easily understand the whole details, although the data are rarely discussed in detail. The authors just list the data in the paper and no useful conclusion has been drawn. Hence, this manuscript is not appropriate for publication in Molecules. Some minor questions:
1) The determination of the diastereomers employing Riguera’s method should be discussed in detail. The results are not that obvious without looking carefully at the 1H NMR spectra. It will be much easier to read if the authors summarize the relative chemical shifts in a table.
2) The chemical shifts and integrals should be given in the 1H NMR spectra. The chemical shifts should be given in the 13C NMR spectra. The ascription of the protons and carbons should be noted clearly in either 1D or 2D spectra, by attaching the structural formula with labeled protons and carbons in the spectra for people to read easily.
3) The data of LC-MS Analysis of - (+)-7-Nitropropranolol and - (-)-7-Nitropropranolol are the same. Please check.
Reviewer 2 Report
This paper describes an efficient methodology to access bioactive and important chiral molecules efficiently. For this reason, it deserves to be published, once more information will be included in the introduction regarding other methods of preparation of title compounds.
Author Response
We really want to thank the Reviewer for his/her good evaluation of our work and for his/her encouraging comments. We added the required details in the introduction section, and we found that our manuscript was further improved.
Reviewer 3 Report
In this manuscript, the authors report ‘Synthesis, Chiral Resolution and Enantiomers Absolute Configuration of 4-Nitropropranolol and 7- Nitropropranolol’.
My specific remarks are presented below: 1. Abstract Although a different precursors were used, the synthetic approach to these kind of compounds is developed elsewhere. Authors just used known procedure, they did not develop it, as they have stated in this abstract. 2. Introduction The Introduction section has to provide a more detailed insight into the previously reported derivatives of propranolol (or propranolol itself) and possible or known biological activity. Only by fast looking the literature avaliable on internet I found several derivatives and their facile synthesis. Authors should include it into Introduction. First section in Introduction is not properly referenced, i.e. [1] reference does not describe 'several studies about....', yet it describes synthesis! 3. Results Since authors describe simple synthesis, which has been previously described in other papers (reference 1), there is no need to frame names of products in such a big box (Sheme 1). It only takes place on Sheme. It could be smaller and bellow the structure of product in one line. Same remark for Sheme 2. 4. Materials and Methods Good work in synthetic way has been done. In section 3.4. (Lines 227-236, 238-246, 256-265, 267-276,284-293,295-303, 311-320,322-331) I don’t see a purpose of writing the synthesis of every MPA-propranolol apart. All sentences are the same, with only difference in mass of compound and mass of EDC, HRESI-MS. Procedure should be written in one part as general procedure, with table of results (necessary reactants and product) below. In every mentioned line, the NMR shifts of chloroform are noted, 7.26 ppm and 77.0 ppm, why? Put it after 126 line, together with purification, and notification about HSQC and HMBC. 5. Conclusion My main concerns are related to the absence of the goal of this work. There are so many experiments that could be done to show importance of synthesis of these compounds. In line 25 authors suggest that nitroaromatic compounds are resistant to oxidative degradation. They could perform any test in comparison with propranolol to deduce how NO2 influence activity, since propranolol is known as drug. Also, they could determine lipophilicity (by simple HPLC method) of derived compounds to deduce their possible biopharmaceutical and therapeutic usage.For all reasons listed before, I suggest authors to revise this manuscript in accordance to these comments. They should offer the use of these derived compounds, either by defining their lipophilicity or other bilogical activity (antioxidant, antimicrobial, oxidative degradation....). It would improve their manuscript making sythesized compounds more suitable for application in biopharmaceutical and therapeutic usage.

Round 2
Reviewer 1 Report
The manuscript has been modified accordingly and I recommend publication as it is.
Author Response
Reviewer: 1
- The manuscript has been modified accordingly and I recommend publication as it is.
We thank the Reviewer for his/her evaluation and the time spent on the re-evaluation of our manuscript.
Reviewer 3 Report
By putting this sentence 'With the aim to investigate if drugs could be subject to the same nitration process, we synthesized 4-nitro and 7-nitropropranolol as probes to evaluate the nitration of the propranolol by the endothelium.' in Abstract, authors suggest that these work (experiment) has been done in this work.
Also, in Introduction they put the same sentence 'With the aim to investigate if drugs could be subject to the same nitration process discovered for endogenous catecholamines, we synthesized 4-nitro and 7-nitropropranolol as probes to evaluate the nitration of the propranolol by the endothelium' misleading the reader that this work has been done.
Please, reformulate this sentence.
I still think that this work needs to be improved with some experiment showing the use of nitration and experimental work that authors have done. The authors did not revise this manuscript in accordance to these comments. Rather, they will consider these comments in their future work.
However, I think that the manuscript requires the mentioned data to obtain a reliable conclusion on use of these compound, not their future use.
Author Response
Reviewers’ comments and Authors’ answers
Reviewer: 1
- The manuscript has been modified accordingly and I recommend publication as it is.
We thank the Reviewer for his/her evaluation and the time spent on the re-evaluation of our manuscript.
Reviewer: 3
We thank the reviewer for his/her evaluation and the time spent on the re-evaluation of our manuscript. We have modified the manuscript according to the suggestions indicated by reviewer.
1) “With the aim to investigate if drugs could be subject to the same nitration process, we synthesized 4-nitro and 7-nitropropranolol as probes to evaluate the nitration of the propranolol by the endothelium.' in Abstract, authors suggest that these work (experiment) has been done in this work. Also, in Introduction they put the same sentence 'With the aim to investigate if drugs could be subject to the same nitration process discovered for endogenous catecholamines, we synthesized 4-nitro and 7-nitropropranolol as probes to evaluate the nitration of the propranolol by the endothelium' misleading the reader that this work has been done. Please, reformulate this sentence.”
- In compliance with the referee's request, we have rephrased the text as follows:
Abstract – lines 14-16 – “In order to investigate whether drugs could be subject to the same nitration process, we synthesized 4-nitro and 7-nitropropranolol as probes to evaluate the possible nitration of the propranolol by the endothelium.”
Introduction - lines 57-61 – “In order to investigate whether drugs could be subject to the same nitration process discovered for endogenous catecholamines, we synthesized 4-nitro and 7-nitropropranolol. These propranolol derivatives will be used as probes to evaluate the possible nitration of the propranolol by the endothelium. Since enantiomers …..”
2) “I still think that this work needs to be improved with some experiment showing the use of nitration and experimental work that authors have done. The authors did not revise this manuscript in accordance to these comments. Rather, they will consider these comments in their future work.
However, I think that the manuscript requires the mentioned data to obtain a reliable conclusion on use of these compound, not their future use.”
We agree with the reviewer that manuscript could be improved by adding pharmacological data. However, the inclusion of both in vivo and in vitro pharmacological data would imply in a new manuscript. Since reviewers #1 and #2 have already considered this version of the manuscript suitable for publication, we believe that the pharmacological data could be subject of a separate publication.